# How Positive and Negative Environmental Behaviours Influence Sustainable Tourism Intentions

**Li-Yao Shien, Chih-Hsing Liu \* and Yi-Min Li**

Department of Tourism Management, National Kaohsiung University of Science and Technology, Kaohsiung City 807, Taiwan; k0932799257@gmail.com (L.-Y.S.); ymli@nkust.edu.tw (Y.-M.L.)
\* Correspondence: phd20110909@gmail.com

**Abstract:** This study developed and examined a theoretical model of moderated mediation in which positive and negative environmental behaviours (e.g., attitudes, destruction, conservation, and eco-friendliness) serve as a moderating mechanism that explains the link between the two critical mediating effects of escape and sustainable experiences on revisit intentions. The results of a study of 483 foreign tourists provide support for our hypothesized model. First, the results showed that motivations have indirect and positive effects on revisit intentions through sustainable experiences and escape-seeking. Second, the moderating effects of positive environmental behaviours were found to be positive, while negative environmental behaviours had negative effects on the dimensions of escape and experience on revisit intentions for sustainable tourism. Third, we discussed how this interesting pattern of the moderated mediation setting could be explained by using the theoretical background and considering previous studies on sustainable tourism.

**Keywords:** environmental behaviour; motivation; escape; experience; revisit intention





## 1. Introduction

The growth of sustainable tourism in Taiwan over the last decade has been substantial [1,2]. This development has been mirrored in sustainability, particularly in tourists' environmentally responsible behaviours (ERB) [3–9]. In previous studies, it is commonly acknowledged that there have been few studies on the nature of exposed foreign tourists, their motivations, their sustainable experiences, and their travel intentions as well as how they can be effectively segmented. This becomes increasingly important if tourism managers or the tourism government aim to focus their activities on sustainability to maximize their potential benefits in sustainable tourism attractions and long-term relationships with international visitors and if operators or associated tourism agencies aim to focus on foreign tourists and implement appropriate promotional strategies or improvements in the international brand equity of the destination image [10]. Moreover, the successful development of sustainable tourism not only increases the country's visibility in attracting domestic and foreign tourists [11,12] and helps to maintain its original and natural resources [13]; maintaining the sustainability concepts of tourism destination development also has noteworthy educational value for new generations [14]. Therefore, travel in areas with rich biodiversity and high environmental consciousness has become a new relaxation and tourism activity and provides the best samples for sustainable tourism education [15]. Tourism managers must pay more attention to tourists' requirements to enhance sustainable activities by encouraging their travel motivations and intentions and by adjusting marketing strategies, operation processes, and resource allocations to fit customers' sustainability requirements [16]. However, until now, few researchers have specifically examined how sustainable motivations, experiences and environmental behaviours influence the intentions for sustainable tourism with a sample of foreign tourists in sustainable tourism or its subsequent performance.

According to the strategy of sustainable tourism development in 2020, Taiwan's tourism market will follow the key trends of globalization of the growing Asian market and the globalization effect as well as localization, cultural relevance, and digital transformation. Taiwan contains abundant natural resources, with forests covering 59% and mountains covering 74% of the country to create the mountainous forested ecosystem [17]. In addition to abundant natural environmental resources, the different climatic conditions generate diverse forests and species, which provide suitable conditions for sustainable tourism [2,15]. Despite the profound contributions of sustainable tourism as a new tourism activity and a great benefit for the development of Taiwan's tourism industry, the question of how environmental behaviours influence foreign tourists' travel intentions remains unanswered. Ref. [18] proposed that people's pro-environmental values change their environmental behaviours and generate utilitarian benefits (tourism). Therefore, without fully considering the context of tourists' attitudes and environmental behaviours in their relationships within sustainability, further contemplation of these critical attributes in the travel intentions of sustainable tourism is needed.

Sustainable tourism is becoming a new trend in world travel. The concepts of environmental conservation knowledge and personal environmental attitudes are keys to promoting sustainability; thus, integrating environmental knowledge, environmental sensitivity, and place attachment into changes in tourists' environmental behaviours has increasingly become a managerial priority [4]. To promote sustainable tourism, researchers have argued that motivations and subjective well-being attitudes are critical attributes in identifying nature-based tourism intentions [19]. Indeed, due to global climate change, customer awareness, and a changeable industrial environment, tourism organizations have catered to tourist needs and have sought a balance between environmental preservation and organizational performance [20]. Specifically, tourism organizations must deliver environmental preservation knowledge and encourage sustainability intentions to customers, tourists, competitors, and suppliers to promote sustainability, especially since the Taiwanese government has implemented a number of developmental projects to promote sustainability [1]. Earlier studies have noted that in the initial stages of encouraging sustainable tourism, tourists' understanding of sustainability and whether it affects consumption is widely acknowledged as a foundational attribute of environmental behavioural changes [21]. However, little research has been conducted in this area to examine how motivational encouragement and sequences influence foreign tourists' sustainable tourism intentions through sustainable experiences and escape-seeking social capital accumulation. Accordingly, this study addresses this subject by considering these interrelationships and how they impact sustainable tourism intentions from the perspective of foreign tourists and the Taiwanese context. Tourism seeks to enhance tourists' realistic findings and their search for authenticity and untouched places in the world; thus, psychological forces of experience and the need to escape are pull motives that are responsible for tourists' choice of a destination [22].

From the sustainable tourism perspective, experiences that connect tourists' value orientations to sustainability increase their willingness to travel [23], and escaping crowds of people provides an important motive for nature-based recreation visitors [24]. However, focusing only on sustainable tourism attributes does not provide good explanations of tourists' motivations or intentions towards travel decisions [1]. Instead, the management of internal and external attributes of sustainable motivations and environmental behaviours to integrate and transform sustainability knowledge into tourists' behavioural intentions is paramount in enhancing the understanding of tourists' sustainable intentions [25]. To integrate different concepts from previous studies, this study focuses on the effects of the relationship between sustainability motivations and tourists' accumulated experiences. Furthermore, this study investigates how tourism experiences can be exploited and transformed to improve foreign tourists' overall travel intentions. Although tourists' sustainable tourism intentions may be influenced by pro-social and environmental attitudes [26], the present study focuses on foreign tourists' motivations and travel decision formulation pro-

cesses, which are essential to understanding the future trends of the tourism industry. This study also adds new considerations for environmental behavioural benefits and impacts that reflect the current status of sustainable tourism implications in Taiwan.

## 2. Literature Review and Hypothesis Development

Past studies on the development of sustainable tourism through nature-based tourism have highlighted the motivations of visitors as significant indicators of sustainable practices, such as learning, attitudes, educational orientations, and environmental behaviours [27]. Most previous research involving sustainability attitudes has asserted that tourists' motivations influence the development of their sustainable tourism experience and affect their psychological identification with specific places [19,28]. Visitors are aware of their push and pull motivations and whether previous memories or experiences can motivate them [29], especially when making travel plans and decisions. The deep experience of developmental stimulation often involves motivation as a prerequisite of emotional tension [1], which influences the intention to visit or willingness to pay a premium for sustainability or eco-tourism [30]. If tourists have pleasurable memories of previous experiences, they may gradually develop the intention to repeat the experience and become familiar with the destination [31]. This can result in repeated intentions for sustainable visitation and good-will towards the sustainability of the destination [19]. Based on this, this study proposes the following:

**Hypothesis 1.** *Motivations have indirect and positive effects on revisit intentions through sustainable experiences.*

In the current context, we also propose that motivations may influence tourists' intentions to engage in sustainable tourism through the need to overcome negative moods and leave a stressful social environment, which is a function of the motivation towards sustainable tourism. Existing studies provide strong support for this presumption by demonstrating a positive relationship between tourists' motivation, escape-seeking, and tourism intentions for environmentally responsible tourism [1]. Specifically, the findings indicate that tourists escape mundane life by seeking to fulfil their psychological and biological needs through sustainable tourism [15]. Nostalgia triggers individuals' memories that are embedded in the ecological environment, thereby allowing tourists to escape their everyday lives [32]. This concept of escape-seeking that drives sustainable travel allows the present study to adopt a widely accepted model of tourism motivation as its theoretical basis in addition to the foundational attributes of a typology of factors related to sustainability. Ref. [28] stated that motivations are an important element of collective actions for sustainable tourism that may help individuals escape daily life, potentially changing their environmentally responsible behaviours and educating and enlightening them. This idea is integrated with the abovementioned concepts that address tourists' social psychological need for escape-seeking. This phenomenon implies that once individuals' motivations are encouraged and change their cognitive processes, individuals will develop certain attitudes towards sustainability. Individuals' motivations are psychologically internalized in terms of feeling release from a stressful social environment. This feeling is affected by cognition as well as the emotional desire to escape daily life. In turn, these motivations connect to an individual's intentions to revisit sustainable tourism.

**Hypothesis 2.** *Motivations have indirect and positive effects on revisit intentions through escape-seeking.*

Environmental behaviours refer to actual actions towards natural environmental concerns or accomplishments by individuals or groups [4]. The measurement of environmental behaviours involves the degree of concern regarding the physical environment and reflects how individuals take action or responsibility for environmental preservation [33]. Individuals with positive environmental attitudes who take action for envi-

ronmental protection satisfy their ethical goals and need for personal uniqueness [34,35]. These sustainable behaviours make their travel actions and experiences more beneficial for self-improvement [36]. Many studies have suggested that environmentally responsible behaviours are an important activator between sustainable experiences and travel intentions. For example, [37] assert that tourists' perceptions of ecotourism experiences influence their moral interest in the environment. Ref. [18] also state that acting green is related to individuals' self-concept, moral norms, and positive feelings and that it maximizes personal utility through sustainability actions. Thus, this study argues that when foreign tourists have positive environmental behaviours and take action for long-term environmental development planning and environmental sustainability, they will receive the enjoyable benefits of escape-seeking and the experiences that sustainable tourism provides because this action makes people feel good. Thus, individuals will be more willing to act and develop environmentally protective behaviours or think about environmental issues when making travel decisions, leading them to display active and responsible behaviours with regard to their revisit intentions.

**Hypothesis 3a.** *Positive environmental behaviours have positive and moderating effects on the relationship between sustainable experiences and revisit intentions.*

**Hypothesis 3b.** *Positive environmental behaviours have positive and moderating effects on the relationship between escape and revisit intentions.*

Negative environmental behaviours refer to tourists' actions that do not contribute to environmental protection, do not acknowledge tourism services as public goods, and do not consider personal benefits as well as the actions of tourists who are less concerned about resource conservation [30]. Personal values and benefits influence tourists' intentions to take action towards pro-environmental tourism consumption [16,21,38], although limited empirical evidence exists regarding this phenomenon. Ref. [30] proposed that higher-order individual values lead to the formation of money-oriented evaluations of sustainability and the prioritization of lower-order needs in environmental protection. In other words, lower-order needs in environmentally protective behaviours reduce the benefits of sustainable experiences and escape-seeking and influence tourists' intentions towards sustainable tourism. Ref. [39] asserted that the overdevelopment of economic and irresponsible behaviours for environmental resource consumption is a key element in destination images that influence foreign tourists' visits.

**Hypothesis 4a.** *Negative environmental behaviours have negative and moderating effects on the relationship between sustainable experiences and revisit intentions.*

**Hypothesis 4b.** *Negative environmental behaviours have negative and moderating effects on the relationship between escape and revisit intentions.*

### 3. Methodology

*3.1. Sample Selection and Data Collection*

This study followed previous studies in selecting tourists in ecological conservation areas to participate in a sustainable tourism survey [15] since these visitors have high environmental consciousness and conservation awareness. To reflect foreign tourists' perspectives on sustainable tourism, the ecological conservation area provides a good setting to examine sustainable experience [40,41]. Therefore, this study selected several ecological conservation areas (Table 1), including Yangmingshan National Park, Zhishan Culture and Ecology Green Park, Jiufen Scenery and the Beitou Hot Spring Museum, as sample collection areas to highlight tourists' sustainability experience based on the work of other Taiwanese researchers [42,43].

**Table 1.** Sample selection from Taiwan conservation area.

| Conservation Area | Description |
|---|---|
| Yangmingshan National Park | It is one of the natural conservation areas in Taiwan with a natural ecosystem as well as wildlife protection. There are many internationally famous hot spots, such as Liuhuanggu (硫磺谷), Zhuzihu (竹子湖), Menghuan Pond (夢幻湖), and the Qingtiangang Grassland Trails (擎天崗). With the characteristics of different subtropical and warm temperate climate zones, this area contains numerous interesting plants and wildlife that attract international tourists. |
| Zhishan Culture and Ecology Green Park | The park includes numerous old fragrant trees, shrines, and lakes. Tourists can overlook the Shuangxi River and have a 360-degree sweeping panoramic view when they make their way to the top of the mountain. Visitors in the park can closely observe tree frogs and other animals. In other words, visitors can have a fully embedded experience within an environmental model of sustainability. |
| Jiufen Scenery | Founded during the Qing Dynasty, this park is located atop a mountain and offers stunning views of the Pacific Ocean. This area was developed when gold was discovered and declined in size when gold mining was discontinued. Jiufen is now an increasingly popular international and local tourist destination because of the quaint old streets, tea houses, and stunning views. Visitors can easily access the Historic Commercial District, Songde Park (頌德公園), and Wufan Tunnel (五番坑道). Visitors can also go hiking on Mount Jilong (基隆山) to view the entire Neihu District, Taipei City, Keelung and the northeast coast seascape. Visitors can easily enjoy sustainable tourism. |
| Beitou Hot Spring Museum | This park was built in 1931 by the Japanese government when it ruled Taiwan. The style and structure of the museum is different from modern buildings. The whole building was built with red bricks and wooden weatherboards. The first floor of the museum demonstrates the history of the Beitou hot spring development. The second floor of the exhibition area has the theme of Beitou industry and the Ketagalan family of "Northern Indigenous People". The rich historical culture merges with the ecological system at the Beitou Hot Spring Water Park, forming an ecological and historical culture and attracting foreign tourists to explore the territory. |

As [44] indicated, structural equation modelling (SEM) is based on an asymptotic statistical examination process in which smaller sample sizes (N = 50) may result in greater variance and standard deviations of the factor correlations of the measured constructs may result in inadequate convergence, inappropriate explanations, and poor results. Therefore, Ref. [44] suggested that a sample size of 200 or more observations provides more support and robustness for the parameter estimation. In this study, a total of 514 questionnaires were distributed to foreign tourists with purposive sampling techniques. After discarding 31 questionnaires that included incomplete or useless answers, 483 questionnaires were retrieved for advanced analysis, resulting in a valid retrieval rate of 93.97%. The demographic information of the foreign tourists is shown in Table 2.

**Table 2.** Demographic profile of the foreign tourists (N = 483).

| Item | N | Percentage | Item | N | Percentage |
|---|---|---|---|---|---|
| *Gender* | | | *Occupation* | | |
| Male | 246 | 50.93% | Student | 86 | 17.81% |
| Female | 237 | 49.07% | Government staff | 60 | 12.42% |
| Sum | 483 | 100% | Agroforestry | 7 | 1.45% |
| | | | Manufacturing | 46 | 9.52% |
| *Level of Education* | | | Business/Services | 160 | 33.13% |
| Junior High School | 18 | 3.73% | Housekeeper/Retirees | 52 | 10.77% |
| Senior High School | 46 | 9.52% | Information Industry | 38 | 7.87% |
| University | 303 | 62.73% | Other | 34 | 7.03% |
| MBA or above | 116 | 24.02% | Sum | 483 | 100% |
| Sum | 483 | 100% | | | |

**Table 2.** *Cont.*

| Item | N | Percentage | Item | N | Percentage |
|---|---|---|---|---|---|
| | | | *Nationality* | | |
| *Age* | | | China | 165 | 34.16% |
| Below 20 | 19 | 3.93% | Japan | 50 | 10.35% |
| 21~30 | 214 | 44.32% | American/Canadian | 96 | 19.88% |
| 31~40 | 102 | 21.12% | Europe | 122 | 25.26% |
| 41~50 | 58 | 12.01% | Australia/New Zealand | 20 | 4.14% |
| Above 51 | 90 | 18.63% | Other | 30 | 6.21% |
| Sum | 483 | 100% | Sum | 483 | 100% |
| *Time of visiting* | | | *Type of Travel* | | |
| One | 322 | 66.67% | Group travel | 157 | 32.51% |
| Second | 46 | 9.52% | Backpacking | 150 | 31.06% |
| Third or above | 115 | 23.81% | Other | 176 | 36.44% |
| Sum | 483 | 100% | Sum | 483 | 100% |

*3.2. Measuring Variable Design*

Table 3 shows the basic statistics of validity and reliability for the items on revisit intentions, motivations, experiences, escape, and positive environmental behaviours. The first variable, "Revisit Intention", primarily referred to the sustainable tourism intention scale that was established and summarized by [45] and included three items to measure foreign tourists' intentions to revisit Taiwan in the near future. The second variable, "Motivation", was established with five items from [27]. These items refer to the motivation to return to nature, meet new people, learn more about nature, remain physically fit and enhance family and friends' affinities. The third variable, "Experience", refers to the excitement, enjoyment, and memorable experiences that sustainable tourism may provide. The five related items were developed by [37]. Fourth, the concept of "Escape" refers to the fact that tourists may experience a change in pace from their everyday life, may escape from their normal stressful social environment, and may overcome negative moods through sustainable tourism. Four items were used to measure "Escape", which were proposed by [30]. Fifth, "Positive environmental behaviours" refers to helping other tourists learn about the preservation of the environment, avoiding pollution or destruction, and helping to maintain the local environmental quality. These behaviours were measured with three items proposed by [46]. All the questions were measured using a 7-point Likert scale from 1 (strongly disagree) to 7 (strongly agree). The reliability (Cronbach's $\alpha$) of each scale was as follows: Revisit intention ($\alpha = 0.819$), Motivation ($\alpha = 0.858$), Experience ($\alpha = 0.864$), Escape ($\alpha = 0.898$), and Positive environmental behaviours ($\alpha = 0.748$). These numbers indicate that the reliability of the measured variables was quite high.

In addition, four sets of "Negative environmental behaviour" items were measured using a 7-point Likert scale from 1 (strongly disagree) to 7 (strongly agree) from different scales of attitudes, destruction, conservation, and eco-friendliness adopted from [16,20,47,48], as shown in Table 4. The items that measured "Negative environmental behaviours" in sustainable tourism development had a good factor loading value (>0.5). Cronbach's $\alpha$ was used to measure the reliability of the subdimensions of negative environmental behaviours. The reliability for each variable was as follows: Attitude ($\alpha = 0.804$), Destruction ($\alpha = 0.849$), Conservation ($\alpha = 0.836$) and Eco-Friendliness ($\alpha = 0.871$).

The final two columns of Tables 3 and 4 show the values of the CR and the AVE that measure the conservative indicator of validity. The values for the variables were as follows: revisit intentions (CR = 0.823; AVE = 0.609), motivations (CR = 0.852; AVE = 0.538), experience (CR = 0.866; AVE = 0.565), escape (CR = 0.899; AVE = 0.692), positive environmental behaviours (CR = 0.767; AVE = 0.532), attitudes (CR = 0.848; AVE = 0.655), destruction (CR = 0.860; AVE = 0.555), conservation (CR = 0.937; AVE = 0.833) and eco-friendliness (CR = 0.879; AVE = 0.595). The CRs were all above the threshold value of 0.7 and the AVEs

had values of 0.5 (or higher), which proves the convergent validity and high proportion of common variance [49].

**Table 3.** Factor loading, average variance extracted (AVE) and composite reliability (CR) for measuring the items of revisit intentions, motivations, experiences, escape, and positive environmental behaviours.

| Measurement Items | Factor Loading | AVE | CR |
|---|---|---|---|
| *Revisit Intentions* | | | |
| 1. I would like to revisit Taiwan in the near future. | 0.745 | | |
| 2. If had to decide again, I would choose Taiwan. | 0.755 | 0.609 | 0.823 |
| 3. I would come back to Taiwan in the future. | 0.837 | | |
| *Motivations* | | | |
| 1. Sustainable tourism provides the opportunity to meet new people. | 0.630 | | |
| 2. Sustainable tourism lets me feel as if I returned to nature. | 0.729 | | |
| 3. Sustainable tourism lets me learn more about nature. | 0.869 | 0.538 | 0.852 |
| 4. Sustainable tourism lets me keep physically fit. | 0.756 | | |
| 5. Sustainable tourism can enhance family and friends' affinities. | 0.661 | | |
| *Experiences* | | | |
| 1. Sustainable tourism provides excitement, enjoyment, and memorable experiences. | 0.742 | | |
| 2. Sustainable tourism lets me feel like I am part of the process to see wildlife and to fulfill wants and needs. | 0.794 | | |
| 3. Sustainable tourism lets me experience new, unique, and different experiences. | 0.699 | 0.565 | 0.866 |
| 4. Sustainable tourism provides the ability to experience the peaceful tranquillity of the natural environment. | 0.820 | | |
| 5. Sustainable tourism provides positive interactions between guests and the lodge staff, guides, and group members. | 0.696 | | |
| *Escape* | | | |
| 1. I take part in sustainable tourism to get away from my normal environment. | 0.890 | | |
| 2. I take part in sustainable tourism to have a change in pace from my everyday life. | 0.912 | 0.692 | 0.899 |
| 3. I take part in sustainable tourism to overcome a bad mood. | 0.721 | | |
| 4. I take part in sustainable tourism to get away from a stressful social environment. | 0.791 | | |
| *Positive environmental behaviours* | | | |
| 1. I will express my opinion to local administration if I find the phenomenon of environmental pollution or destruction. | 0.581 | | |
| 2. I will actively help tourists to learn about sustainable tourism. | 0.896 | 0.532 | 0.767 |
| 3. I will help to maintain the local environmental quality. | 0.675 | | |

**Table 4.** Factor loading, average variance extracted (AVE) and composite reliability (CR) for measuring items of negative environmental behaviours.

| Measurement Items | Factor Loading | AVE | CR |
|---|---|---|---|
| *Negative Environmental Behaviours* | | | |
| *Attitudes* | | | |
| 1. In our country, we have enough electricity, water, and trees that we do not have to worry about conservation. | 0.842 | | |
| 2. The earth is a closed system where everything eventually returns to normal, so I see no need to worry about its present state. | 0.930 | 0.655 | 0.848 |
| 3. Keeping separate piles of garbage for recycling is too much trouble. | 0.626 | | |
| *Destruction* | | | |
| 1. Sustainable tourism is destroying environmental quality. | 0.869 | | |
| 2. Sustainable tourism is destroying the local environment. | 0.742 | | |
| 3. Sustainable tourism is causing the loss of traditional culture. | 0.615 | 0.555 | 0.860 |
| 4. Sustainable tourism is destroying public security. | 0.738 | | |
| 5. Sustainable tourism is increasing congestion in peak periods. | 0.738 | | |
| *Conservation* | | | |
| 1. I do not want to do everything that I can to protect and conserve wildlife. | 0.873 | | |
| 2. We do not have the responsibility to leave healthy ecosystems for our families and future generations. | 0.945 | 0.833 | 0.937 |
| 3. We do not need to help protect animals and animal habitats. | 0.919 | | |
| *Eco-Friendly* | | | |
| 1. I do not intend to buy environmentally friendly products in the future. | 0.824 | | |
| 2. I do not intend to buy organic food in the future. | 0.834 | | |
| 3. I do not intend to reduce household waste in the future. | 0.807 | 0.595 | 0.879 |
| 4. I do not intend to use products made from recycled material whenever possible. | 0.746 | | |
| 5. I do not intend to avoid genetically modified foods in the future. | 0.625 | | |

*3.3. Common Method Variance (CMV)*

Previous studies have suggested that common method variance may present a potentially serious bias in behavioural research on foreign tourists [10]. Several procedures were adopted to calculate whether the variables that were used in this study involve potential common method bias. The first procedure that was used was the Harman one-factor test, which uses concepts from [50] texts on factor analysis to detect CMV. The principal component analysis revealed that the first factor accounted for less than 50% of the variance among variables and indicated no serious CMV problem.

Recently, tourism researchers have suggested confirmatory factor analysis (CFA) as an additional test to detect CMV. We used AMOS 18.0 [10] and kept the theoretical basis in perspective when detecting causal connections between variables. The results of this CFA revealed a good fit of the observed data to the measurement model ($\chi^2_{(575)}$ = 1147.841; CFI = 0.942; AGFI = 0.865; GFI = 0.883; IFI = 0.943; RMSEA = 0.045). Based on the multiple statistical tests for detecting CMV, as shown in Table 5, the results can be interpreted as indicating that common-method variance may not be a serious problem or threat in this study.

**Table 5.** Environmental behaviour measurement model: CFA goodness-of-fit statistics.

| Indicator of CFA (Chekalina et al., 2018) | Suggest Value | Statistic Value |
|---|---|---|
| Normed fit index (**NFI**) | >0.90 | 0.892 |
| Relative fit index (**RFI**) | >0.90 | 0.881 |
| Incremental fit index (**IFI**) | >0.90 | 0.943 |
| Comparative fit index (**CFI**) | >0.90 | 0.942 |
| Goodness of fit index (**GFI**) | >0.90 | 0.883 |
| Adjusted goodness of fit index (**AGFI**) | >0.80 | 0.865 |
| Root mean square error of approximation (**RMSEA**) | <0.08 | 0.047 |
| Chi-Square ($\chi^2$)/degree of freedom | <3.0 | 2.052 |

## 4. Results

Table 6 shows the descriptive statistics of all the variables. The correlations among some variables were represented by high values, indicating sufficient variation and an advanced examination process to identify the multicollinearity effects. For example, the correlation of negative environmental behaviours of destruction and conservation was 0.628. The values of VIF indicate that multicollinearity is not a serious concern in our observation variables since all the factors were below 2.05 [10].

**Table 6.** Means, standard deviations (SD), variance inflation factors (VIF), intercorrelations, and correlation matrixes among variables.

| Variables | Mean | S.D. | 1. | 2. | 3. | 4. | 5. | 6. | 7. | 8. | 9. | VIF |
|---|---|---|---|---|---|---|---|---|---|---|---|---|
| 1. Revisit Intention | 5.142 | 1.040 | (0.819) | | | | | | | | | |
| 2. Motivation | 6.099 | 0.933 | 0.222 *** | (0.858) | | | | | | | | 2.05 |
| 3. Experience | 5.229 | 1.024 | 0.417 *** | 0.116 * | (0.864) | | | | | | | 1.88 |
| 4. Escape | 5.250 | 1.372 | 0.182 *** | 0.493 *** | 0.096 * | (0.898) | | | | | | 1.73 |
| 5. Positive Behavioral | 4.915 | 1.139 | 0.339 *** | 0.115 * | 0.254 *** | 0.087 | (0.748) | | | | | 1.51 |
| *Negative Behavioral* | | | | | | | | | | | | |
| 6. Attitude | 2.298 | 1.314 | −0.106 * | −0.316 *** | 0.013 | −0.187 *** | −0.150 *** | (0.804) | | | | 1.43 |
| 7. Destroy | 1.759 | 1.175 | −0.051 | −0.304 *** | 0.061 | −0.153 *** | −0.147 ** | 0.462 *** | (0.849) | | | 1.34 |
| 8. Conservation | 1.830 | 1.056 | −0.095 * | −0.287 *** | 0.062 | −0.085 | −0.171 *** | 0.391 *** | 0.628 *** | (0.836) | | 1.12 |
| 9. Eco-Friendly | 2.023 | 1.242 | −0.105 * | −0.275 *** | 0.066 | −0.155 *** | −0.112 * | 0.444 *** | 0.505 *** | 0.594 *** | (0.871) | 1.11 |

Note. N = 483 hotel managers. The internal consistency reliabilities of alpha coefficients are shown on the diagonal in bold. * $p < 0.05$, ** $p < 0.01$, *** $p < 0.001$.

The hypothesized model has a good fit to the data, as represented by the following statistics: $\chi^2$ (114, N = 483) = 238.116, $p < 0.001$, RMSEA = 0.048, CFI = 0.971, AGFI = 0.926, GFI = 0.945, and IFI = 0.971. The standardized coefficients are shown in Figure 1.

Hypothesis 1 asserted that motivations have indirect and positive effects on revisit intentions through sustainable experiences. As shown in Figure 1, motivations were positively related to experiences ($\beta = 0.152$, $p < 0.01$), and experiences were positively related to revisit intentions ($\beta = 0.454$, $p < 0.001$). Furthermore, the indirect effect of motivations on revisit intentions was $\beta = 0.069$, $p < 0.05$, which supports Hypothesis 1. The results also support Hypothesis 2. The direct effect of motivations was positively related to escape ($\beta = 0.536$, $p < 0.001$) and escape was positively related to revisit intentions ($\beta = 0.121$, $p < 0.01$), which suggested that motivations indirectly affected revisit intentions through escape ($\beta = 0.065$, $p < 0.05$).

To further assess the overall model fit of the mediating effects of Hypothesis 1 and Hypothesis 2, alternative overall models were tested that added a direct path from motivations to revisit intentions (alternative model). As shown in Table 7, although the alternative model also shows good model fit ($\chi^2_{(113)}$ = 315.348; CFI = 0.953; AGFI = 0.906; GFI = 0.930; IFI = 0.953; RMSEA = 0.061), the original model has better model fit than the alternative model. Therefore, since there was no noticeable improvement in the alternative model, the original hypothesized mediation model was maintained as the superior model because it was the most parsimonious.

The results of Figure 1 support Hypothesis 3a and indicate that the interaction between positive environmental behaviours and sustainable experiences was significantly related to revisit intentions ($\beta = 0.126$, $p < 0.001$). Figure 2 and the slope tests show that the

relationship between sustainable experiences and revisit intentions was stronger when positive environmental behaviours were high rather than low.

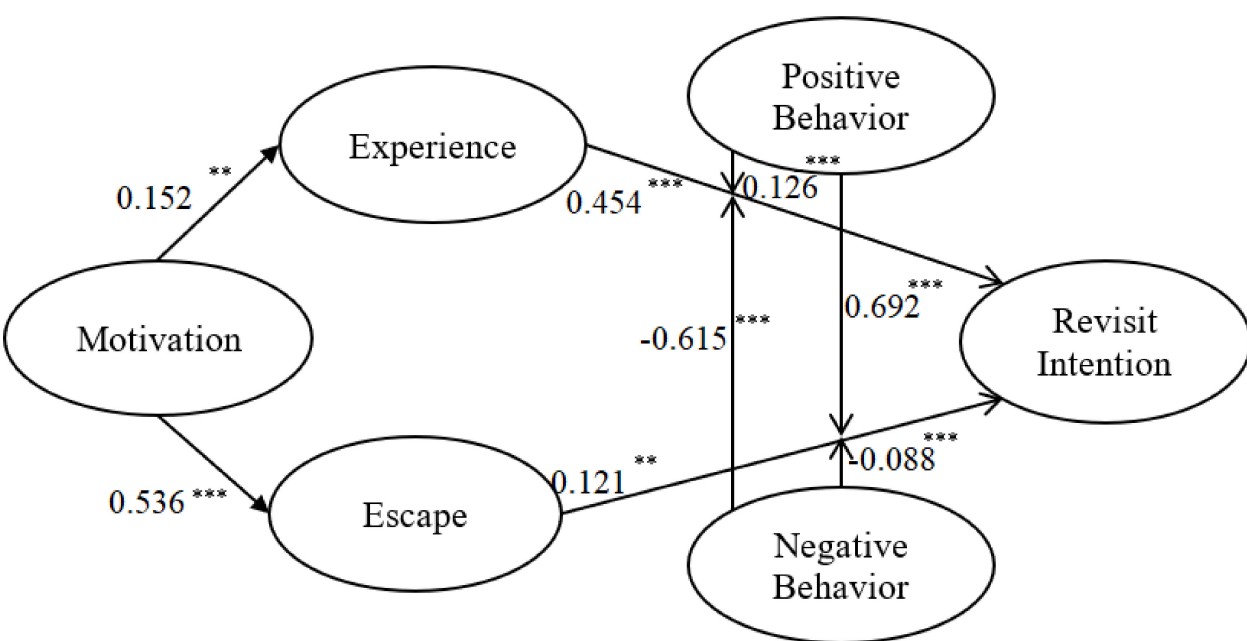

**Figure 1.** Structural Model and Standardized Paths with Study Variables. ** $p < 0.01$, *** $p < 0.001$.

**Table 7.** Comparison of competing models (predicting new service development).

| Model Test | $\chi^2$ | *df* | $\chi^2/df$ | *p*-Value | CFI | AGFI | GFI | IFI | RMSEA |
|---|---|---|---|---|---|---|---|---|---|
| Hypothesized Model (indirect effect) | 238.116 | 114 | 2.089 | 000 | 0.971 | 0.926 | 0.945 | 0.971 | 0.048 |
| Alternative Model 1 (add path from Motivation to Revisit Intention) | 315.348 | 113 | 2.766 | 000 | 0.953 | 0.906 | 0.930 | 0.953 | 0.061 |

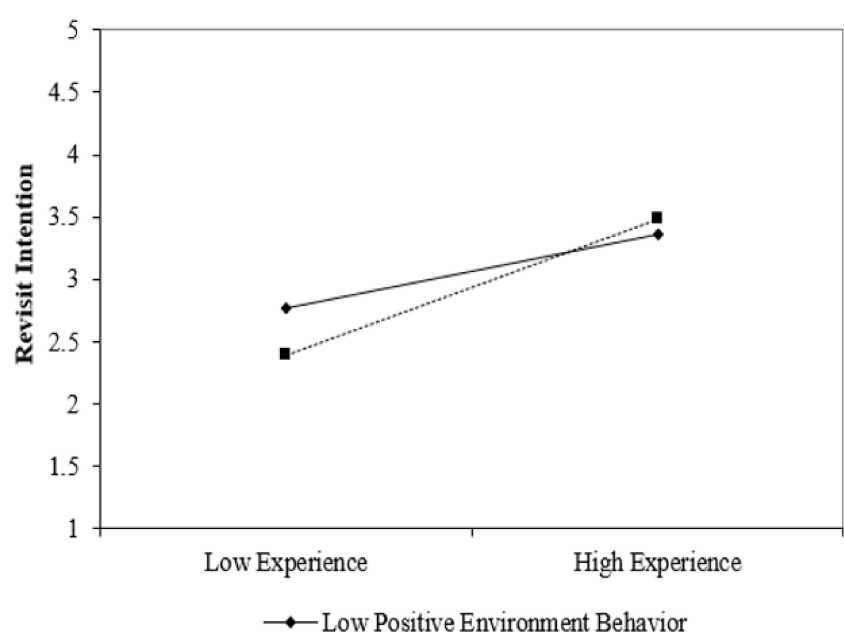

**Figure 2.** Interaction of Positive Environmental Behaviours and Experiences on Revisit Intentions.

In support of Hypothesis 3b, the coefficient for the interaction effects among positive environmental behaviours and escape on revisit intentions was significant ($\beta$ = 0.692, $p$ < 0.001). In the comparison results, shown in Figure 3, the slope difference for the interaction relationship when foreign tourists' positive environmental behaviours and escape were high in revisit intentions (rather than low) was significant. Thus, the findings support Hypothesis 3b.

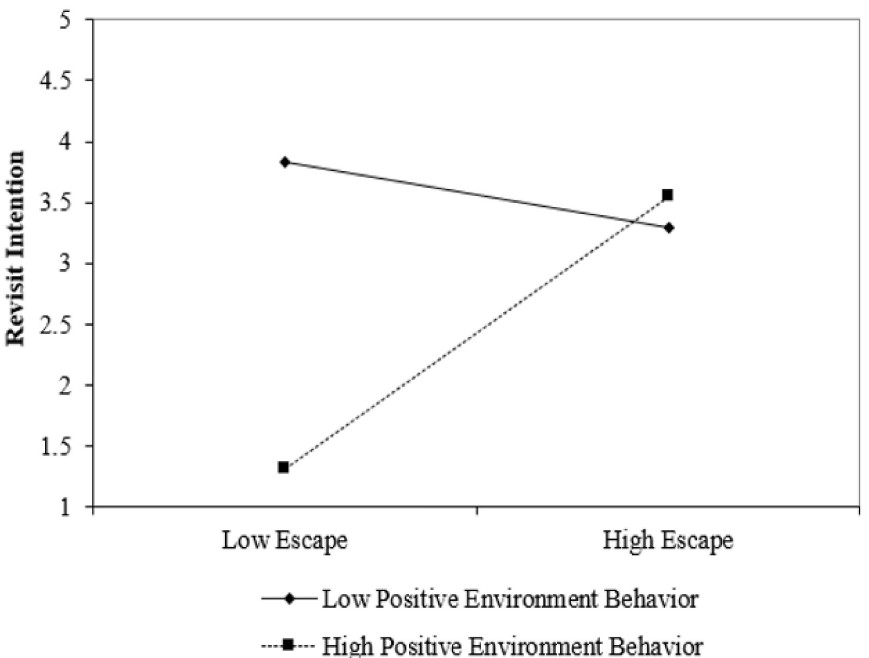

**Figure 3.** Interaction of Positive Environmental Behaviours and Escape on Revisit Intention.

In addition, the results of Figure 1 support Hypothesis 4a and indicate that the interaction between negative environmental behaviours and sustainable experiences was significantly related to revisit intentions ($\beta$ = −0.615, $p$ < 0.001). Figure 4 and the slope tests show that the relationship between sustainable experiences and revisit intentions was weaker when negative environmental behaviours were low rather than high.

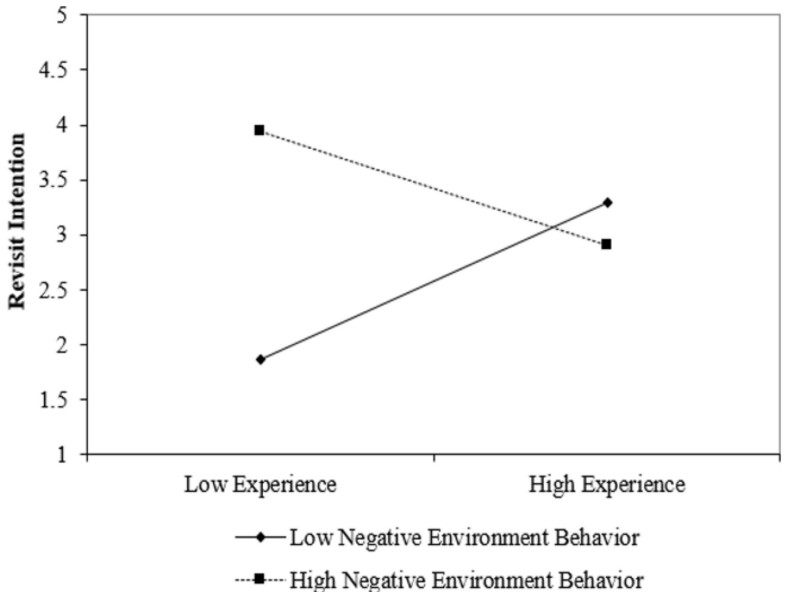

**Figure 4.** Interaction of Negative Environmental Behaviours and Experiences on Revisit Intention.

In support of Hypothesis 4b, the coefficient for the interaction effects among negative environmental behaviours and escape on revisit intentions was significant ($\beta = -0.088$, $p < 0.001$). In the comparison results shown in Figure 5, the slope difference for the interactive relationship when foreign tourists' negative environmental behaviours and escape were low in revisit intentions versus when foreign tourists' negative environmental behaviours and escape were high in revisit intentions was significant. Thus, the findings support Hypothesis 4b.

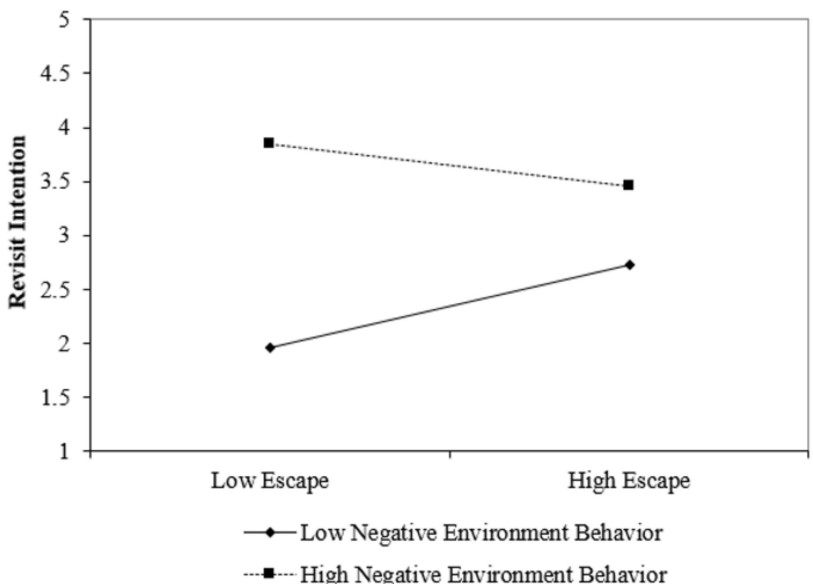

**Figure 5.** Interaction of Negative Environmental Behaviours and Escape on Revisit Intentions.

## 5. Discussion and Conclusions

Sustainable concepts and behavioural development are core issues for future tourism and hospitality activity. This development will allow visitors' awareness of environmental sustainability to create a natural atmosphere that meets the needs of consumers [25,37]. Prior tourism and hospitality studies have identified numerous challenges faced by sustainable tourism when confronted with high economic developmental conflicts and environmental conservation [51]. Specifically, in studies on sustainable tourism development, Ref. [4] found that tourists' ERBs were the top challenges faced by sustainable island tourism development. Consequently, the current research identified the critical features of sustainable tourism by focusing on how foreign tourists' motivation, escape-seeking, and experience of developmental processing impact their revisit intentions when they have different environmental behaviours and awareness. First, this study extended the study by [19] by asserting that tourists' motivations are effective predictors of their subjective well-being in nature-based tourism and identified multiple mediating results that indicated that escape and experience are critical mediating roles that link foreign tourists' motivations and revisit intentions. The results show that tourists' values and sustainability needs are the foundational basis of environmental sustainability, and they reflect the concept that "tourists' satisfaction and needs are key for sustainable tourism development" [1]. Therefore, any sustainable knowledge transfer or creative experience accumulation activities in a tourism organization or governmental department must focus on tourists' motivations and inspiration to enhance their sustainable lifestyle and must focus on the diffusion of the responsibility for the negative consequences of tourism [21].

Second, the results show that visitors' pro-environmental behaviours trigger the enhancement of tourists' sustainable experiences since they connect the personal benefits of acting green and targeting individual economic interests, which can lead to better environmentally sustainable behaviours and revisit intentions in the future [18]. This finding follows [19] argument that environmental behaviours contribute to the development

of nature-based tourism by playing important roles in influencing individuals' travel decisions. Importantly, this study reveals that negative environmental behaviours not only impact the destination image [39] but also result in varying intentions of visiting among foreign tourists. To control the potential impacts on or benefits of sustainable tourism intentions, positive and negative environmental behaviours are critical because they are associated with levels of ecological knowledge and conservation sensitivity, which influence tourists' affection towards specific destinations [4]. Tourism and hospitality organizations might create educational environments to inspire tourists' motivations that can support unforgettable experiences and escape-seeking and improve tourists' respect and affection for sustainability. This can encourage organizations to develop new services or products in response to customers' need for sustainability. As a result, the contributions of environmental sustainability, tourist satisfaction, and organizational performance can be achieved.

### 5.1. Theoretical Contributions

The present study fills the gaps in the previous literature on sustainable tourism and highlights several theoretical contributions. First, in studies of sustainable tourism for foreign tourists' motivations and revisit intentions in the Taiwanese context, previous researchers have suggested that tourists' motivation is an important contextual factor that can influence travel intentions and decisions [1,52]. However, sustainability issues in the Taiwanese context have not been well documented, especially with respect to the consideration of foreign tourists' perspectives. Following the increased attention to environmental conservation in Taiwan's tourism and hospitality markets to attract foreign tourists, the exploration of foreign tourists' viewpoints on sustainable tourism not only contributes to the literature but also provides meaningful implications for marketing strategies and the allocation of limited tourism resources [10]. This study aims to extend the literature by identifying foreign tourists' motivations and experiences in relation to sustainable tourism intentions. The findings may provide insight to tourism and hospitality studies into how this empirical evidence could be useful for the intangible attributes of sustainable experiences.

Second, at a broad level, previous sustainability studies have strongly suggested that environmental behaviours and attitudes play key roles in determining payment premiums and revisit intentions for sustainable tourism [10,30]. We believe that the present research is the first to provide a cohesive explanation and empirical evidence for the relationship between positive and negative environmental behaviours and revisit intentions. In addition, this research illustrates how specific environmental behaviours influence the sustainable tourism decision-making process. Furthermore, this work explains how this influence is exerted on sustainable intentions through the encouragement of positive environmental behaviours and their impacts on negative environmental behaviours. In fact, motivations, experiences, and environmental behaviours constitute three instrumental determinants of tourists' travel intentions. They belong to a different research stream that, remarkably, has not previously been systematically integrated as a means to explain foreign tourists' sustainability behaviours and tourism intentions.

Third, this study uses the well-developed behavioural theory of different environmental attitudes for a sustainability–intention model in a mediation-moderation setting. This is especially true with the introduction of the second-order factor analysis of negative environmental behaviour with regard to attitudes, destruction, conservation, and eco-friendliness. Past researchers [47] have used this perspective on behavioural analysis to predict consumers' eco-friendly intentions, but few studies have examined foreign tourists' behaviours and perspectives based on these ideas or based on advanced mediation-moderation analysis.

Fourth, by extending tourists' motivations and perspectives and incorporating the sustainable experience developmental process into sustainable intentions [21], this study shows that foreign tourists' revisit intentions for sustainable tourism can be encouraged depending on their motivations, which are inspired by the experiences and escape mechanisms that sustainable tourism may provide. This paper also extends the tourism and

hospitality literature with empirical tests of theories of "sustained value creation" for environments [53]. Future tourism managers should be more careful about ethical objectives from the perspective of tourists, especially when sustainability is connected with destination resource conservation and protection [54]. However, most sustainable tourism studies in Taiwan only focus on environmentally responsible behaviours and on the effects of community participation [15] and do not consider how to develop and encourage foreign tourists' motivations. Studies should be conducted with a more comprehensive environmental behavioural analysis that considers the complex and dynamic tourism environment to meet customers' requirements and changing needs for sustainability [10].

### 5.2. Managerial Implications

Several implications are highlighted for sustainable tourism management. First, to satisfy the sustainability requirements of customers' needs, tourism organizations must take responsibility for the environment, community, and ecosystem and should develop "'new tourism" to meet customers' changing demands and to survive in the increasingly complex and dynamic environment [53]. In the highly competitive tourism environment of Taiwan, because of limited resource support from the government and the regulatory constraints of environmental protection laws, managers are required to address customers' sustainability needs and to create sustainable tourism environments to attract tourists and maintain organizational growth and survival [55]. Ref. [34] proposed that managers need to consider the demand-side approach to increase customers' knowledge and awareness of environmentally friendly ways to decrease environmentally harmful behaviours.

Additionally, this study emphasizes the benefits of tourists' sustainable experiences and escape-seeking when engaging in sustainable tourism. This concept is similar to the "effective supply-side approach", which refers to changing the services provided by changing tourists' travel behaviours with regard to environmental consequences and requiring firms to satisfy tourists' expectations for minimum standards of service provision [34]. In other words, managers must identify the necessity of sustainable experiences for tourists because tourism organizations can benefit from increased organizational revenue when tourists participate in sustainable activities [56]. In addition, managers can contribute to the objectives of sustainable development as citizens of the earth [57]. Furthermore, managers need to identify the potential influences of negative environmental behaviours, consider the importance of sustainable experiences and escape-seeking, and increase tourists' awareness of global environmental changes. Consequently, managers must enhance their responses to tourists' changes in requirements and improve tourists' intentions to revisit.

Finally, with respect to the marketing strategy formulation process, it is important to understand customers' needs and future tourism trends before implementing resource allocation and marketing strategies. Given the finding that encouraging tourists' sustainable motivation can increase their overall travel intentions through experiences and escape-seeking, managers who aim to encourage tourists' sustainable motivations should consider establishing an education system to develop tourists' environmental knowledge, environmental sensitivity, and place attachment to attract them and inspire their sustainable tourism intentions [4].

### 5.3. Limitations and Future Research Directions

Despite the theoretical and empirical contributions to sustainable tourism development, this study also has several limitations and makes suggestions for future work on sustainable tourism. First, this study sampled foreign tourists and considered their sustainable experiences and perspectives to increase the representativeness and to reflect the real tourism market and requirements. It is possible that the study of real sustainable tourism phenomena should consider foreign tourists' opinions outside of a particular location and should include the opinions and perspectives of insiders, such as stakeholders [58], domestic tourists [1,46], or residents [59]. In addition, the creation and integration of value-added sustainable products or services for tourists should be considered. As [60]

suggested recently with regard to developing customers' sustainable intentions, hospitality and tourism firms' supply chain management (SCM) should empower customers to become codesigners, coproducers, and comarketers of sustainability services and should increase customers' consumption intentions to achieve sustainability. Hence, future sustainable tourism research should extend the findings of this study and examine the generalizability of the present findings by sampling different participants or sustainability service providers across a range of SCM. This should include the perceptions of stakeholders, domestic tourists or residents, which could increase opportunities for sustainable products and services and deliver them to tourists to make operations more sustainable.

In addition, although the theoretical model that was used in this study included the system analysis process and integrated diverse methods of sustainable tourism that assessed new trends in sustainability research, the key variables were measured within a short time frame. A longitudinal survey could observe changes in sustainability concepts, or a cross-sectional design could be used. This could include experienced and high-potential tourism organization managers or owners since they may be more willing to take on challenging jobs or may use alternative methods of problem-solving, which could make profound contributions to organizational and sustainable tourism development [52]. Therefore, a useful next step for future research would be to apply a cross-sectional design to sustainability development that responds to environmental changes and contributes to overall organizational performance.

Finally, environmental behaviours are depicted in this study as external attributes that can influence sustainable experiences and improve escape-seeking. However, there are other external attributes that can have negative or positive moderating effects in different contexts. For example, regarding environmental behaviours, it has been suggested that environmental knowledge, environmental sensitivity, and place attachment should be considered together when examining the effects of the environmentally responsible behaviours of tourists [4,55]. Thus, exploring the various moderating roles of other external attributes of environmental conservation concepts would be a valuable direction and would have more implications for future sustainable tourism research.

**Author Contributions:** Conceptualization, L.-Y.S. and C.-H.L.; methodology, C.-H.L.; software, Y.-M.L.; validation, L.-Y.S. and C.-H.L.; formal analysis, C.-H.L.; investigation, Y.-M.L.; resources, Y.-M.L.; data curation, C.-H.L.; writing—original draft preparation, C.-H.L.; writing—review and editing, C.-H.L. and Y.-M.L. All authors have read and agreed to the published version of the manuscript.

**Funding:** This research received no external funding.

**Institutional Review Board Statement:** Not applicable for this studies because of not involving humans or animals.

**Informed Consent Statement:** Informed consent was obtained from all subjects involved in the study.

**Conflicts of Interest:** The authors declare no conflict of interest.

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
