# Peer review of "How Positive and Negative Environmental Behaviours Influence Sustainable Tourism Intentions"

_sustainability, doi:10.3390/su14116922_

Round 1

Reviewer 1 Report

Overall the study is interesting; however, the draft needs proofreading and editing. I suggest Authors should avail English editing service.

Author Response

Overall the study is interesting; however, the draft needs proofreading and editing. I suggest Authors should avail English editing service.

Thanks for your suggestions. Based on your suggestions, we have asked two non-experts to go through the first part of the manuscript and have sent the revised manuscript to American Journal Experts (AJE) to help us to correct the grammar, sentencing and paraphrasing. The American Journal Experts (AJE) website is http://www.journalexperts.com/en/. AJE has forged partnerships with hundreds of journals, including many titles published by Nature and Elsevier. If the edited version of this paper still cannot reach your high quality requirement, we will change the editing association. We have already disclosed this problem to AJE. We have also attached the certificate letter that we received from AJE.

Reviewer 2 Report

The authors developed and analyzed a theoretical model to study the positive and negative environmental behaviors of sustainable tourism.  They used research with foreign tourists to validate the model. They used appropriate statistical methods. The article is very interesting. The research hypotheses were properly verified. I only have a few comments. These are intended to make the article clearer. 

row. 331 The standardized coefficients are shown in the paths of Figure 1. 

How it was calculated standardized coefficients?

How it was calculated: 

the indirect effect of motivations on revisit intentions is β = .069, p < .05

motivations would indirectly affect revisit intentions through escape (β = .065, p < .05). 

Figure 1 is not very clear to me. 

  1. arrows suggest that one variable is dependent and the other independent
  2. the arrows have different graphic forms, does this suggest anything?  
  3. there are the links between positive and negative behaviour 

In general, the figure is not clear and does not give more than a table 

The conclusions are correct and interesting 

Author Response

The authors developed and analyzed a theoretical model to study the positive and negative environmental behaviors of sustainable tourism.  They used research with foreign tourists to validate the model. They used appropriate statistical methods. The article is very interesting. The research hypotheses were properly verified. I only have a few comments. These are intended to make the article clearer. 

Thank you very much for your time and effort in working with us on this paper. We are grateful for the comments that you have provided. We have followed your instructions and made the appropriate changes. We hope you will find the revised paper more clear and compelling than the earlier version. The point-by-point concerns you raised regarding the copy editing are addressed below.

row. 331 The standardized coefficients are shown in the paths of Figure 1. 

How it was calculated standardized coefficients?

How it was calculated

the indirect effect of motivations on revisit intentions is β = .069, p < .05

motivations would indirectly affect revisit intentions through escape (β = .065, p < .05).

Thank you for your comments. The values of the standardized coefficients are calculated with AMOS software following previous studies to show the standardized coefficients and the results of direct or indirect effects (Horng et al., 2022; Lee et al., 2022; Liu et al., 2022; Liu & Jiang, 2020). We thank you for your valuable suggestions.

New reference:

Horng, J. S., Liu, C. H., Chou, S. F., Yu, T. Y., & Hu, D. C. (2022). Marketing Management in the Hotel Industry: A Systematic Literature Review by Using Text Mining. Sustainability14(4), 2344.

Lee, W. L., Liu, C. H., & Tseng, T. W. (2022). The multiple effects of service innovation and quality on transitional and electronic word-of-mouth in predicting customer behaviour. Journal of Retailing and Consumer Services64, 102791.

Liu, C. H., Gan, B., Ko, W. H., & Teng, C. C. (2022). Comparison of localized and foreign restaurant brands for consumer behavior prediction. Journal of Retailing and Consumer Services65, 102868.

Liu, C. H., & Jiang, J. F. (2020). Assessing the moderating roles of brand equity, intellectual capital and social capital in Chinese luxury hotels. Journal of Hospitality and Tourism Management43, 139-148.

Figure 1 is not very clear to me. 

  1. arrows suggest that one variable is dependent and the other independent
  2. the arrows have different graphic forms, does this suggest anything?  
  3. there are the links between positive and negative behaviour

In general, the figure is not clear and does not give more than a table 

Thank you for your suggestion. Figure 1 demonstrates the results of the hypothesis. First, as you noted, the arrows suggest that one variable is dependent and the other is independent. The basic model for structural equation modelling (SEM) estimation is shown and may be useful in explaining complex phenomena when predicting tourist behaviour (Horng et al., 2016; Jiang et al., 2017; Liu et al., 2018). Second, the arrows have different graphic forms; does this suggest anything? The negative behaviour variable has four subdimensions. To avoid misleading readers, we have removed the subdimensions of attitude, destruction, conservation and eco-friendliness. Third, positive and negative behaviour are the moderating variables proposed in this study.

Reference:

Horng, J. S., Wang, C. J., Liu, C. H., Chou, S. F., & Tsai, C. Y. (2016). The role of sustainable service innovation in crafting the vision of the hospitality industry. Sustainability8(3), 223.

Jiang, W. H., Li, Y. Q., Liu, C. H., & Chang, Y. P. (2017). Validating a multidimensional perspective of brand equity on motivation, expectation, and behavioural intention: a practical examination of culinary tourism. Asia Pacific Journal of Tourism Research22(5), 524-539.

Liu, C. H., Horng, J. S., Chou, S. F., Huang, Y. C., & Chang, A. Y. (2018). How to create competitive advantage: the moderate role of organizational learning as a link between shared value, dynamic capability, differential strategy, and social capital. Asia Pacific Journal of Tourism Research23(8), 747-764.

The conclusions are correct and interesting 

Thank you for providing such valuable opinions to help us improve the overall quality of our research. Overall, your suggestions are insightful and very helpful for revising our paper. We believe that you will find the paper has substantially improved. Your comments have been very valuable in highlighting basic weaknesses in the paper’s positioning. We have taken each comment very seriously and tried to do a better job of developing the underlying theoretical concerns and highlighting the research gaps that shape this research. Thanks again, and have a nice day!

Reviewer 3 Report

This study developed and examined a theoretical model of moderated mediation in which positive and negative environmental behaviors (e.g., attitudes, destruction, conservation and eco-friendliness) serve as a moderating mechanism, which explains the link between the two critical mediating effects of escape and sustainable experiences in revisit intentions.
The article is interesting and well structured. I suggest these additions to improve its quality:
- the abstract should be made more detailed in the description of the results
- bibliographic references should be extended and updated. I suggest citing this article: Marinello et al., 2021. Indicators for sustainable touristic destinations: a critical review
- to what extent are the sites described in Table 1 representative of Taiwan's conservation areas?

Author Response

This study developed and examined a theoretical model of moderated mediation in which positive and negative environmental behaviors (e.g., attitudes, destruction, conservation and eco-friendliness) serve as a moderating mechanism, which explains the link between the two critical mediating effects of escape and sustainable experiences in revisit intentions.
The article is interesting and well structured. I suggest these additions to improve its quality:

Thank you very much for your time and effort in working with us on this paper. We are grateful for the comments that you have provided. We have followed your instructions and made the appropriate changes. We hope you will find the revised paper more clear and compelling than the earlier version. The point-by-point concerns you raised regarding the copy editing are addressed below.

- the abstract should be made more detailed in the description of the results

Thank you for your suggestion. We have rewritten the “abstract” in the revised manuscript.

Original abstract: This study developed and examined a theoretical model of moderated mediation in which positive and negative environmental behaviours (e.g., attitudes, destruction, conservation and eco-friendliness) serve as a moderating mechanism, which explains the link between the two critical mediating effects of escape and sustainable experiences in revisit intentions. The results of a study of 483 foreign tourists provide support for our hypothesized model. As expected, the moderating effects of positive environmental behaviours are found to be positive effects, while negative environmental behaviours have negative effects on the dimensions of escape and experience on revisit intentions for sustainable tourism. Alternatively, we also found that motivation influences revisit intentions through escape and sustainable experiences. We discuss how this interesting pattern of the moderated mediation setting could be explained by using the theoretical background as well as discussing previous studies on sustainable tourism.

New abstract: This study developed and examined a theoretical model of moderated mediation in which positive and negative environmental behaviours (e.g., attitudes, destruction, conservation and eco-friendliness) serve as a moderating mechanism that explains the link between the two critical mediating effects of escape and sustainable experiences on revisit intentions. The results of a study of 483 foreign tourists provide support for our hypothesized model. First, the results show that motivations have indirect and positive effects on revisit intentions through sustainable experiences and escape-seeking. Second, the moderating effects of positive environmental behaviours are found to be positive, while negative environmental behaviours have negative effects on the dimensions of escape and experience on revisit intentions for sustainable tourism. Third, we discuss how this interesting pattern of the moderated mediation setting could be explained by using the theoretical background and considering previous studies on sustainable tourism.

- bibliographic references should be extended and updated. I suggest citing this article: Marinello et al., 2021. Indicators for sustainable touristic destinations: a critical review

Thank you for your suggestion. We have included the Marinello et al., 2021 in the revised manuscript.

Original sentence: Moreover, the successful development of sustainable tourism not only increases the country’s visibility in attracting foreign tourists and helps to maintain its original and natural resources, but it also strengthens the sustainability for tourism industry development and has noteworthy educational sense for new generations.

New sentence: Moreover, the successful development of sustainable tourism not only increases the country’s visibility in attracting domestic and foreign tourists (Cogato et al., 2019; Marinello et al., 2020) and helps to maintain its original and natural resources (Marinello et al., 2021); maintaining the sustainability concepts of tourism destination development also has noteworthy educational value for new generations (Borsato et al., 2018).

New reference:

Borsato, E., Galindo, A., Tarolli, P., Sartori, L., & Marinello, F. (2018). Evaluation of the grey water footprint comparing the indirect effects of different agricultural practices. Sustainability10(11), 3992.

Cogato, A., Meggio, F., De Antoni Migliorati, M., & Marinello, F. (2019). Extreme weather events in agriculture: A systematic review. Sustainability11(9), 2547.

Marinello, S., Lolli, F., & Gamberini, R. (2020). The impact of the COVID-19 emergency on local vehicular traffic and its consequences for the environment: The case of the city of Reggio Emilia (Italy). Sustainability13(1), 118.

Marinello, S., Andretta, M., Lucialli, P., Pollini, E., & Righi, S. (2021). A Methodology for Designing Short-Term Stationary Air Quality Campaigns with Mobile Laboratories Using Different Possible Allocation Criteria. Sustainability13(13), 7481.

- to what extent are the sites described in Table 1 representative of Taiwan's conservation areas?

Thank you for your comments. We have added more explanation and studies in the revised manuscript.

Original sentence: To highlight the foreign tourists’ perspectives on sustainable tourism, several ecological conservation area descriptions were included as follows: Yangmingshan National Park, Zhishan Culture and Ecology Green Park, Jiufen Scenery and the Beitou Hot Spring Museum (Table 1).

New sentence: To reflect foreign tourists’ perspectives on sustainable tourism, the ecological conservation area provides a good setting to examine sustainable experience (Gavin et al., 2018; Hosen et al., 2020). Therefore, this study selected several ecological conservation areas (Table 1), including Yangmingshan National Park, Zhishan Culture and Ecology Green Park, Jiufen Scenery and the Beitou Hot Spring Museum, as sample collection areas to highlight tourists’ sustainability experience based on the work of other Taiwanese researchers (Huang & Sun, 2019; Liu & Tien, 2019).

New reference:

Gavin, M. C., McCarter, J., Berkes, F., Mead, A. T. P., Sterling, E. J., Tang, R., & Turner, N. J. (2018). Effective biodiversity conservation requires dynamic, pluralistic, partnership-based approaches. Sustainability10(6), 1846.

Hosen, N., Nakamura, H., & Hamzah, A. (2020). Adaptation to climate change: Does traditional ecological knowledge hold the key?. Sustainability12(2), 676.

Huang, S. C. L., & Sun, W. E. (2019). Exploration of social media for observing improper tourist behaviors in a national park. Sustainability11(6), 1637.

Liu, T. M., & Tien, C. M. (2019). Assessing Tourists’ Preferences of Negative Externalities of Environmental Management Programs: A Case Study on Invasive Species in Shei-Pa National Park, Taiwan. Sustainability11(10), 2953.